# Fostering a Clean and Sustainable Environment through Green Product Purchasing Behavior: Insights from Malaysian Consumers' Perspective

**Nabil Hasan Al-Kumaim** [1,*], **Muhammad Salman Shabbir** [2], **Salman Alfarisi** [1], **Siti Hasnah Hassan** [3], **Abdulsalam K. Alhazmi** [4], **Sanil S. Hishan** [5], **Samer Al-Shami** [1], **Nadhmi A. Gazem** [6], **Fathey Mohammed** [7] and **Hussein Mohammed Abu Al-Rejal** [8]

1   Faculty of Technology Management and Technopreneurship, Universiti Teknikal Malaysia Melaka (UTeM), Melaka 75450, Malaysia; nabil@utem.edu.my (S.A.); samerali@utem.edu.my (S.A.-S.)
2   College of Commerce and Business Administration, Dhofar University, Salalah 211, Oman; mshabbir@du.edu.om
3   School of Management, University Science Malaysia, Penang, Gelugor 11800, Malaysia; siti.hassan@usm.my
4   Faculty of Engineering and Computing, University of Science and Technology, Aden 891, Yemen; a.alhazmi@ust.edu
5   Azman Hashim International Business School, Universiti Teknologi, Kuala Lumpur 54100, Malaysia; sshishan@utm.my
6   Department of Information Systems, College of Business Administration-Yanbu, Taibah University, Medina 42353, Saudi Arabia; nalqub@taibahu.edu.sa
7   School of Computing, University Utara Malaysia, Sintok 06010, Kedah, Malaysia; fathey.mohammed@uum.edu.my
8   School of Technology Management and Logistic, University Utara Malaysia, Sintok 06010, Kedah, Malaysia; abualrejal@uum.edu.my
*   Correspondence: nhs1426@yahoo.com

**Abstract:** Rapid economic developments have led to the excessive consumption of environmental resources. Consumption patterns play a crucial role in deteriorating environmental conditions and influencing consumers to seek sustainability features while purchasing different products. The purpose of this paper was to analyze the sustainability factors that have prompted consumers in Malaysia to buy green products. The primary elements of this research focused on environmental concern, green product awareness, government support, perceived ecological value, community green practice, purchase intention, and green product purchase behavior. Additionally, to explain the relationship between the independent and dependent variables, this research employed the theory of planned behavior as a theoretical framework. A total of 300 questionnaires were collected and examined using smart PLS-SEM. The findings of the research suggest that all factors, including environmental concern, green product awareness, government support, perceived ecological value, community green practice, and purchase intention, influence consumers in Malaysia to purchase green products. Finally, this research discusses the contribution, limitations, and suggestions for future studies related to purchasing behavior towards green products.

**Keywords:** environmental concern; green practice; ecological value; government support; purchase intention; green product purchase behavior

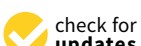

## 1. Introduction

Over the last decade, consumer demand for products and services has risen dramatically, contributing to natural resource depletion and substantial environmental damage. Global warming, increased pollution, and a loss of flora and wildlife are just a few of the devastating effects of environmental destruction [1]. Consequently, recent research has found that resource depletion and environmental damage negatively affect the three

expected sustainability outputs, namely, a firm's productivity, consumer happiness, and community environmental sustainability [2].

In response, companies have introduced environmentally friendly products and services to mitigate the damage caused to the ecosystem. Therefore, firms should take greater responsibility to design greener products [3]. In line with such an approach, purchasing environmentally friendly products with wise consumption actions will help preserve a clean and healthy atmosphere; otherwise, the chance for a sustainable ecosystem will be lowered [4].

The increased focus on environmentally sensitive goods and services means buyers are focusing their buying behavior on green products. The concept of sustainable development has arisen as a result of this awareness and concern for the environment, highlighting the requirement for sustainability and pushing for growth that has the least adverse impact on the environment and society [5,6].

Additionally, according to Chen et al. [7], as several types of environmental damage result from product materials, manufacturing processes should be minimized. Therefore, by purchasing green items, consumers can help to prevent or mitigate environmental damage. While evidence suggests that the total number of consumers keen on purchasing green products has improved recently, there remains little solid proof that sales of green products have risen. In spite of the interest in environmentally friendly goods and constructivist attitudes towards sustainability, the overall market share of green products only accounts for 1–3% of the total market share [8]. This situation means that ecological issues have a minor role in consumer buying choices and that most people remain unaware of the environmental cost of their consumption.

In Malaysia, consumers display high levels of environmental awareness but show moderate outcomes in purchasing green products [9]. Such circumstances prove that some consumers prefer product convention regardless of the harmful effects of conventional products on the environment [10]. According to Anvar et al. [11], the decision not to buy green products causes difficulties for internal marketers in formulating a more efficient green product marketing strategy due to consumer attitudes that place less emphasis on environmental sustainability. Some consumers express a desire to buy green products to protect the environment but tend to forget this intention upon entering stores. This attitude encourages major supermarkets, such as Tesco and Aeon, and the Institute of Standards and Industrial Research Malaysia (SIRIM), to find ways to promote green products, including electronic goods, green packaging items and other types of green products, to change the attitude of consumers and protect nature [12].

In Malaysia, people have difficulty finding green items, which can be expensive in some supermarkets. Evidence suggests that the difficulties in finding stores that sell green products, as well as the products' comparatively high prices, encourage consumers to overlook the benefits of such items [13]. Consequently, marketers have begun to develop green marketing strategies to help consumers find green products and deliver these items at affordable prices. This strategy is likely to raise consumers' intention to purchase green products, as consumers will likely be driven to buy green products that are easy to identify and obtain at a reasonable cost.

Furthermore, a lack of consumer confidence in green products poses a problem for shoppers buying these goods [14]. In Western countries, before making any purchases, consumers will seek advice from friends and family and search for information about green products from mainstream media sources such as the Internet, radio, and television [15]. Recommendations by consumers' social groups, friends, and family not only facilitate household affairs but also save the environment and increase the intention to buy green products [13]. In Malaysia, most consumers do not use green products. This situation provides an opportunity for the government, marketers, and the mass media to attempt to make people aware of the advantages of green products through promotions and campaigns [16]. Therefore, this research aims to identify and analyze factors that foster environmental sustainability and influence consumers in Malaysia to buy green products.

Information extracted from the Scopus database covering 2010–2022, as shown in Figure 1A–C, indicates that only a few studies in Malaysia focus on the sustainability factors that lead consumers to purchase green products. According to Figure 1A, the number of studies in Malaysia related to the keywords "Pollution" AND "Malaysia" exceeded 930. On the other hand, as shown in Figure 1B, the number of studies related to the keywords "Pollution" AND "Malaysia" AND "environment" AND "consumption" exceeded 66. However, the studies in Malaysia related to the factors leading consumers to purchase green produce amounted to less than six. Therefore, this study was initiated to analyze related sustainability factors that influence consumers in Malaysia to buy green products.

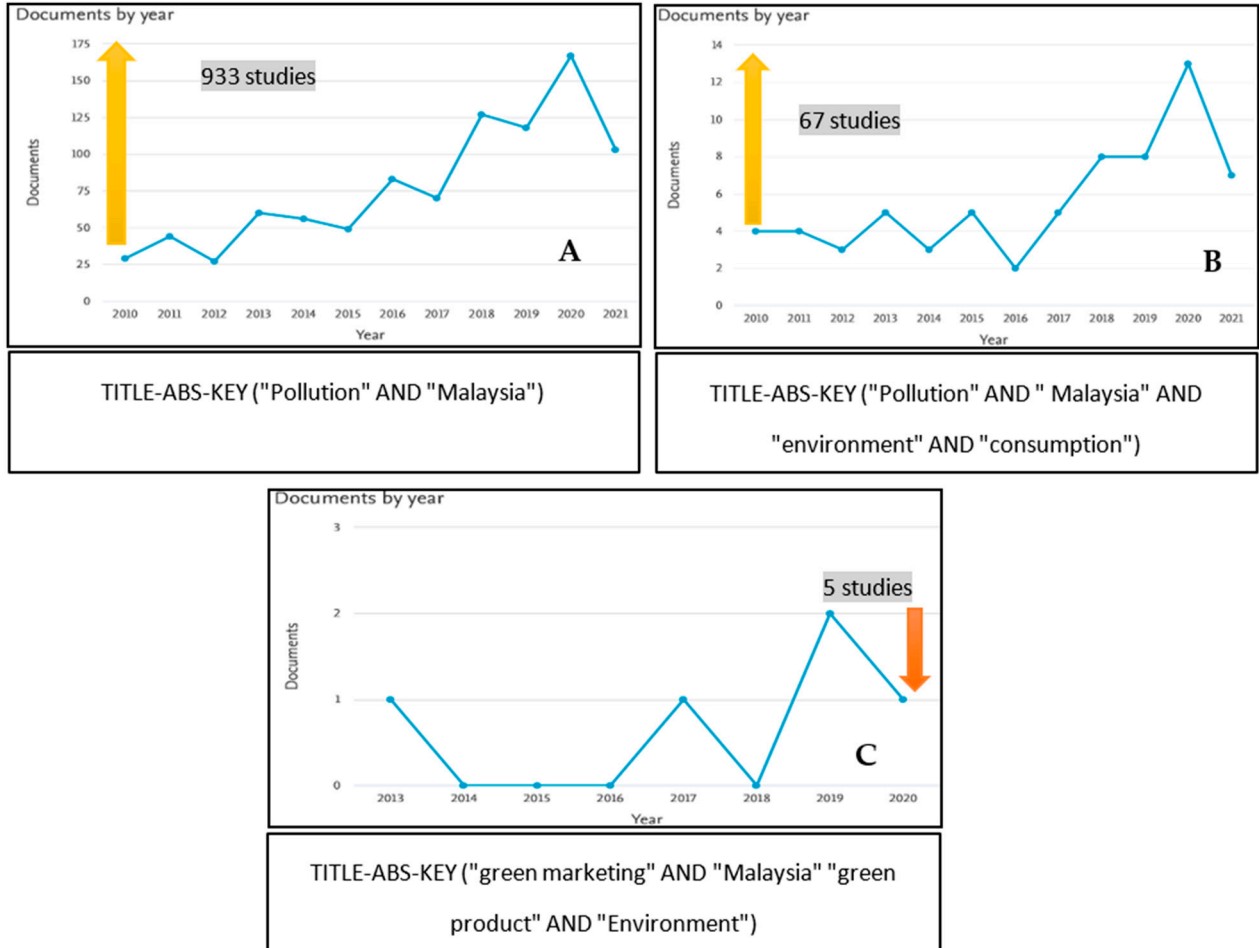

**Figure 1.** Illustration of the number of studies related to pollution, environment, and consumption in Malaysia compared to the number of studies related to green marketing, green products, and the environment.

## 2. Literature Review

### 2.1. Underpinning Theory (Theory of Planned Behavior)

According to Maichum and Parichatnon [16], the central argument of planned behavior theory developed by Ajzen [17] is that human behavior arises from rational decisions rather than intentional action and through its effect on behavioral intentions is affected by mood, subjective norm, and perceived behavioral control. Human behavior is controlled by behavioral motives, which are in turn affected by attitude (target behavior evaluation), subjective standards (judgment on the future attitude of others towards the target behavior), and perceived behavioral influence (the perceived ability to perform the target behavior) [18]. Many studies have agreed that many different behaviors are correctly predicted by the theory of planned behavior (TPB) [19]. Several other studies proposed that more predictors should be applied to the expected behavior theory to increase its capacity to

explain [20]. TPB has been demonstrated to be useful to predict the intention of consumers to purchase green products [21–24]. Therefore, this research employed TPB to predict consumer purchase intention of green product by examining contextual factors of environmental concern, green product awareness, government support, perceived ecological value, and community green practice toward actual green product purchasing behavior.

### 2.2. Hypotheses and Research Framework Development

### 2.2.1. Environmental Concern

Environmental concern includes awareness of ecological issues and the willingness to support and personally contribute to the management of environmental problems [24–26]. The relationship between environmental concern and the purchase of green goods and actions has shown inconsistencies. The vigorous role of consumers represents a way of accomplishing ecological concepts by encouraging environmental conservation. Environmentally conscious consumers do not always act in an ecologically friendly manner [27]. Consumers' willingness to recycle products, care about environmental destruction, and pay for more environmentally friendly products has been reflected. On the other hand, consumers who show less concern about the environment demonstrate less willingness to pay more for green products than those who are more concerned [26,28]. Hence, the first hypothesis is put forward as follows:

**Hypothesis 1 (H1).** *There is a significant relationship between environmental concern and purchase intention towards green product purchase behavior.*

### 2.2.2. Green Product Awareness

Consumer choice behavior for green products is reliant on the level of general awareness of the benefits of green products, which enhances customers' perceptions of the utility obtained from product and encourages them to link a product with a specific group, and raising general understanding of environmental problems [28]. Green product refers to a product that is usually non-toxic, recycled, or reusable, not animal-tested, non-polluting, minimally packaged, and containing natural ingredients, recycled materials, and approved chemicals [29]. When the hazardous material of products became an issue, companies began to develop environmentally friendly or green products, as well as green product policies to influence customer purchasing decisions. Price is the characteristic that customers rely on when making a green buying decision. Customers are less likely to purchase green products if they cost more [30,31]. Without the need to trade off quality and pay higher rates for it, all goods sold should be environmentally friendly. The basic hypotheses is put forward as follows:

**Hypothesis 2 (H2).** *There is a significant relationship between green product awareness and purchase intention towards green product purchase behavior.*

### 2.2.3. Government Support

Government and media reports play a significant role in influencing consumer perceptions and attitudes towards the environment and products. Environmental protection is regarded as one of the primary government responsibilities [31,32]. Consequently, government support helps shape a company's market green culture and sustainable performance and influences consumer orientation towards purchasing green and sustainable products [33,34]. Government policies also predict the attitudes of consumers towards the environment. The government's position in environmental protection positively influences consumer attitudes towards green products [35]. The stance of the government will play a significant role in providing guiding principles to consumers, which, in turn, can affect the buying attitude based on the health and safety values of green products. According to this line of reasoning, the following hypothesis is proposed:

**Hypothesis 3 (H3).** *There is a significant relationship between government support and purchase intention towards green product purchase behavior.*

### 2.2.4. Perceived Ecological Value

Consumers who care about the environment prefer ecologically natural products, such as wooden items from sustainable forests, organic vegetables, ozone-friendly aerosols, biodegradable and non-animal-tested products, and unleaded fuel [36]. According to previous research, the concept of green perceived value comprises the ratings of "green" goods purchased by consumers and comparing the perceived advantages of obtaining the product, which includes the need for environmentally friendly goods [32]. Green perceived value is essential because it can increase the interest of consumers in green product purchases [35]. The perceived value collects characteristics linked to the understanding of how the product value contributes to a clean and safe natural environment. However, the link between consumption and potentially harmful effects on the natural environment has become more crucial than ever [37]. Therefore, purchases of green products and the ecological consumption of their non-green equivalents could result from awareness of such products' perceived environmental value. Hence, the hypothesis is put forward as follows:

**Hypothesis 4 (H4).** *There is a significant relationship between perceived ecological value and purchase intention towards green product purchase behavior.*

### 2.2.5. Community Green Practice

Consumer values, habits, and the purchase of green goods are motivated by environmental and sustainability concerns. People who participate in eco-friendly programs are more likely to purchase green products. For enterprises, pursuing green consumers can be tricky [38]. Not only do consumers want green goods, but they also want businesses to participate in green practices, such as recycling and energy conservation. Reforestation operations, creating plant medicines, garbage recycling to make compost, and reducing field burnings signify green practices that safeguard the environment [39]. Consumers invest in green products even though such goods are more expensive because they contribute to a clean and safe environment. For instance, growing interest in buying electronic devices is a kind of green practice [40]. To sustain their lifestyle, these consumers engage in environmental actions through monetary means. They are likely to spend more money on green products than green consumers who favor environmental legislation and feel they have to solve environmental problems [41]. Hence, the hypothesis is put forward as follows:

**Hypothesis 5 (H5).** *There is a significant relationship between community green practice and purchase intention towards green product purchase behavior.*

### 2.2.6. Green Product Purchase Intention

According to Ajzen [18], purchase intention is influenced by nominated subjective norms (beliefs about what others think about what is right or wrong), behavioral patterns (individuals' beliefs about the outcomes associated with a behavior), and perceived behavioral control (individuals' beliefs about their control over adopting a specific behavior). Thus, buying intention is influenced by consumer attitudes, which affect their perceptions and can drive them to make a certain decision [42].

The aim of buying green goods corresponds to the intention of consumers' desire to purchase a product that causes less harm to the environment and society. Here, the goal is for a buyer to purchase an environmentally friendly product or brand after learning about its green features [36]. According to Vazifehdoust and Taleghani [43], the desire of the consumers to purchase green goods significantly reflects the positive attitude and perceived green value of the products. Green buying intention constitutes an individual's likelihood and ability to prioritize goods that have more eco-friendly features than traditional

products. According to previous studies, one of the primary green purchase intentions correlates to environmental problems. Consequently, the volume of green products has increased and the need for green products is changing the world. Thus, firms are ready to adopt the green production model and introduce it to consumers [44].

One of the emerging areas of study in the broader topic of green marketing is establishing a connection between environmentally friendly consumer attitudes and green purchase intention and behavior [45]. Moreover, understanding green purchase behavior and consumers' attitudes toward environmentally friendly items can benefit businesses looking for insights into sustainable marketing methods [46]. Therefore, the hypothesis is put forward as follows:

**Hypothesis 6 (H6).** *There is a significant relationship between green product purchase intentions and green product purchase behavior. Figure 2 presents the visualization of the research framework.*

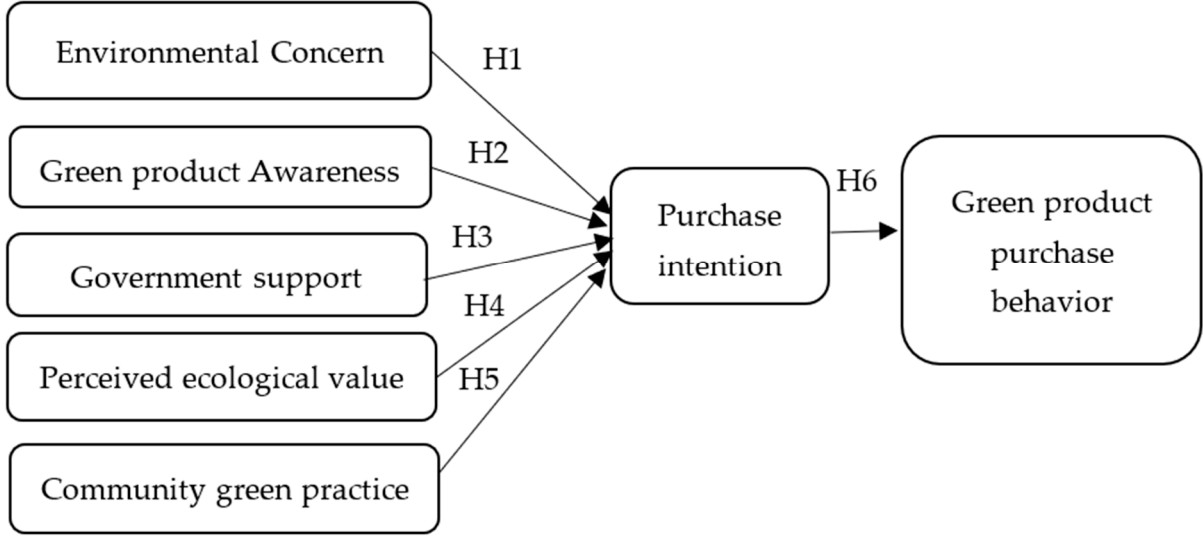

**Figure 2.** A proposed research framework.

## 3. Methodology

### 3.1. Instrument Development, Sampling, and Measures

This research aims to examine the factors that influence consumers in Malaysia to buy green products. A quantitative research methodology gathered data collection and aided analysis. A survey questionnaire acted as a data collection instrument for this research. All constructs were measured using a well-developed multi-item scale adapted from the literature as shown in Table A1 in the Appendix A. Four items adopted from [47,48] measured environmental concern. Green product awareness was measured using four items adapted from [49]. Government support was measured on a four-item scale adapted from [50]. Four items adopted from [50,51] measured perceived ecological value. Community green practice was measured using four items adapted from [52]. Purchase intention was measured using four items adapted from [47,48]. Green product purchase behavior was measured using four items adapted from [48,50]. The Smart PLS3 evaluated and analyzed the data obtained from survey questionnaires.

The research found that the Likert scaling method was adequate to calculate the questionnaire items in this analysis [52]. Using the five-point Likert scale, respondents described the degree to which they agreed or disagreed with each statement (1 = strongly disagree; 2 = disagree; 3 = neutral; 4 = agree, and 5 = strongly agree).

### 3.2. Sampling and Procedures

An online research questionnaire carried out from 1st May to 20th of June 2021. This research adopted a purposeful sampling approach and selected respondents with experience relevant to the factors that influence consumers in Malaysia to buy green products. The choice of this sampling technique reflected potential respondents' prior purchases of green products. The final survey used 300 participants who had experience with purchasing green products. According to Anderson and Gerbing [53], for reasonable estimation, the minimum sample size should comprise 100–150 subjects. However, other authors have suggested a minimum sample size of 200 respondents [54]. The research obtained 300 usable responses; 47.4% were male, and 52.6% were female. Nearly one-third of the respondents fell in the 21–25 age group. The majority of respondents were college or university-educated (76.6%) and had been in employment (87%).

## 4. Data Analysis and Results

To investigate the measurement and structural model, this study employed Partial Least Square-Structural Equation Modeling (PLS-SEM) with SmartPLS v. 3.2.8 [55] as a statistical tool. Such a method is particularly well suited to this work since it allows for smaller sample sizes without depending on normalcy assumptions, which is necessary because survey research is not normally distributed [56]. This study followed Anderson and Gerbing's guidelines and used a two-step strategy to examine the measurement model, followed by a structural model evaluation [57].

### 4.1. Assessment of Measurement Model

All confirmed construct items underwent further examination in the measurement model to see if they contributed significantly to the current research's suggested mode. A measuring model evaluation for this analysis assessed convergent validity, such as outer loading, average variance extracted, and composite reliability. Invalidity discriminations required the employment of cross-loading. Validity evaluation is also crucial for accurate business studies. The average variance extracted (AVE) and the discriminating validity are two types of validity criteria evaluations [58]. Figure 3 shows the measuring model.

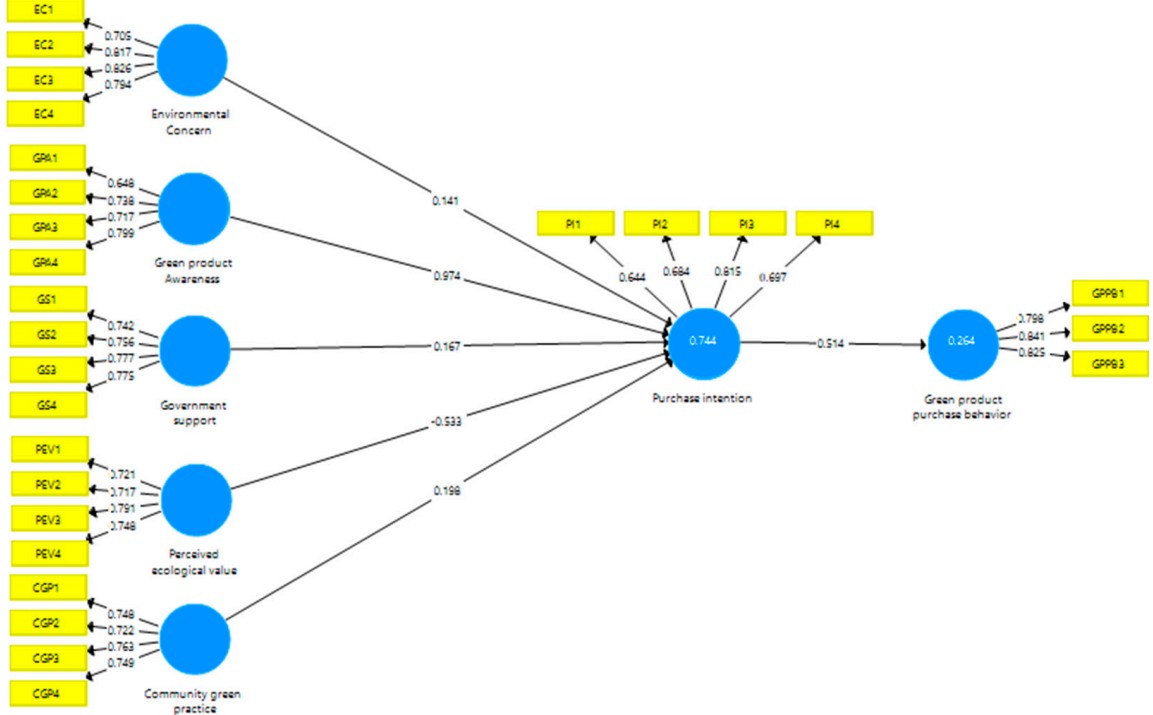

**Figure 3.** Measurement model.

### 4.1.1. Reliability and Convergent Validity

This study used both convergent and discriminant validity to assess the assessment items and constructs. Table 1 lists the tests for reliability and convergent validity. Composite reliability determined reliability, with values higher than 0.7 considered satisfactory [59,60]. The item loadings and average variance extracted (AVE) for each construct check convergent validity, which assesses the degree of items' connection to the construct as theoretically envisioned [59,60]. All item loadings were higher than 0.7, and for AVE, all constructs were higher than 0.50 [61], showing that the measuring model had appropriate convergent validity. Table 1 presents the measurement model outcomes.

**Table 1.** Measurement model.

| Construct | Items | Loadings | Composite Reliability | Average Variance Extracted (AVE) |
|---|---|---|---|---|
| Environmental Concern | EC1<br>EC2<br>EC3<br>EC4 | 0.705<br>0.817<br>0.826<br>0.794 | 0.866 | 0.619 |
| Green Product Awareness | GPA1<br>GPA2<br>GPA3<br>GPA4 | 0.648<br>0.738<br>0.717<br>0.799 | 0.817 | 0.530 |
| Government Support | GS1<br>GS2<br>GS3<br>GS4 | 0.742<br>0.756<br>0.777<br>0.775 | 0.848 | 0.582 |
| Perceived Ecological Value | PEV1<br>PEV2<br>PEV3<br>PEV4 | 0.721<br>0.717<br>0.791<br>0.748 | 0.833 | 0.555 |
| Community Green Practice | CGP1<br>CGP2<br>CGP3<br>CGP4 | 0.748<br>0.722<br>0.763<br>0.749 | 0.833 | 0.556 |
| Purchase Intention | PI1<br>PI2<br>PI3<br>PI4 | 0.644<br>0.684<br>0.815<br>0.697 | 0.803 | 0.508 |
| Green Product Purchase Behavior | GPPB1<br>GPPB2<br>GPPB3 | 0.798<br>0.841<br>0.825 | 0.862 | 0.675 |

### 4.1.2. Discriminant Validity

The degree to which constructs diverge from each other is known as "discriminant validity". Comparing shared variance with all other components with the latent construct's unique variance provides evidence of discriminant validity [61]. The average variance amount in indicator variables that a construct can explain is known as average variance extracted (AVE). According to Fornell and Larcker, the square root of AVE was higher than the squared internal correlations among research constructs. Table 2 indicates the achievement of discriminant validity. Such findings show that bell-shaped normal distribution approximates the data. Both assessments highlight the reliability and validity of the measurement items, thus allowing for hypothesis testing.

**Table 2.** Fornell-Larcker Criterion.

|      | CGP   | EC    | GS    | GPA   | GPPB  | PEV   | PI    |
|------|-------|-------|-------|-------|-------|-------|-------|
| CGP  | 0.746 |       |       |       |       |       |       |
| EC   | 0.415 | 0.787 |       |       |       |       |       |
| GS   | 0.652 | 0.449 | 0.763 |       |       |       |       |
| GPA  | 0.650 | 0.468 | 0.577 | 0.728 |       |       |       |
| GPPB | 0.424 | 0.687 | 0.461 | 0.478 | 0.822 |       |       |
| PEV  | 0.649 | 0.486 | 0.628 | 0.673 | 0.488 | 0.745 |       |
| PI   | 0.652 | 0.495 | 0.585 | 0.669 | 0.514 | 0.618 | 0.713 |

*4.2. Assessment of Structural Model*

After determining that the measurement model or outer models are accurate and valid, the structural model or inner model is assessed. After determining the accuracy and validity of the measurement model or outer models, the structural model or inner model undergoes assessment. Analyzing the model's predictive capacity and the interactions between the components forms part of the process [58]. In other words, structural model assessment evaluates the hypothesized relationship within the internal model. In this research, Figure 2 shows the independent variable, the dependent variable, and the mediating variable in the measurement model. The assumed relationship between paradigms in the current investigations correlates to the following standards:

a   Coefficient of determination ($R^2$) of endogenous constructs,

b   Effect size ($f^2$) and,

c   Path coefficients

4.2.1. Coefficient of Determination ($R^2$)

The $R^2$ value indicates how the endogenous latent variable is impacted by the exogenous construct. In other words, the coefficient represents the amount of variance in endogenous constructs explained by all exogenous constructions linked to it. $R^2$ values below 0.2 are considered weak in academic research, according to studies [62]. Previous research [63] has shown that values of approximately 0.670 are significant, values around 0.333 are normal, and values of 0.190 and below are weak. The R-squares of all endogenous variables in the research model that fall within the moderate prediction are shown in Table 3 and Figure 2.

**Table 3.** Coefficient of determination ($R^2$) of endogenous constructs.

| Construct                     | R Square | R Square Adjusted | Effect      |
|-------------------------------|----------|-------------------|-------------|
| Green product purchase behavior | 0.264    | 0.262             | Moderate    |
| Purchase intention            | 0.744    | 0.740             | Substantial |

4.2.2. Effect Size ($f^2$)

The impact size indicates the relative effect on the endogenous latent variable of a specific exogenous latent variable with R-square adjustments [64]. The external structure, as a static measurement, is critical in explaining the endogenous structure. The effect size ($f^2$) is utilized in analysis to assess whether the removed construct has a substantial impact on the endogenous constructs; the effect size is calculable by comparing the rise in $R^2$ to the percentage of the variance of the remaining unexplained endogenous latent variable. Guidelines indicate that 0.02–0.14, 0.15–0.34, and higher than 0.35 are defined by [65] as minimal, moderate, and high effects, respectively. Table 4 provides the ($f^2$) meaning for each direction.

**Table 4.** Result for effect size ($f^2$).

| Path | $f^2$ | Effect Size |
|---|---|---|
| Community green practice | 0.070 | Low |
| Environmental concern | 0.056 | Low |
| Government support | 0.053 | Low |
| Green product awareness | 0.832 | High |
| Perceived ecological value | 0.234 | Moderate |
| Purchase intention | 0.359 | High |

### 4.2.3. Path Coefficients

The path coefficient is utilized by PLS-SEM to evaluate the power and significance of the latent construct's hypothesized relations. Estimates are derived with a coefficient closer to +1 indicating a strong positive link, a coefficient closer to −1 indicating a strong negative relationship, and structural model associations with uniform values between −1 and +1 [59].

### 4.2.4. Hypothesis Testing

Table 5 shows the hypothesis testing (direct effect). Figure 3 shows the model's path coefficients. The theoretical basis for studying the connection between environmental concern, green product awareness, government support, perceived ecological value, community green practice, purchase intention, and green product purchase behavior is provided by this research model. Figure 4 displays the structural model's output results.

**Table 5.** Hypothesis testing (direct effect).

| Hypothesis | Path | Original Sample (O) | Standard Deviation (STDEV) | T Statistics (|O/STDEV|) | $p$ Values | Results |
|---|---|---|---|---|---|---|
| H1 | EC –> PI | 0.141 | 0.039 | 3.651 | **0.000** | Supported |
| H2 | GPA –> PI | 0.974 | 0.076 | 12.793 | **0.000** | Supported |
| H3 | GS –> PI | 0.167 | 0.052 | 3.208 | **0.001** | Supported |
| H4 | PEV –> PI | −0.533 | 0.078 | 6.851 | **0.000** | Supported |
| H5 | CGP –> PI | 0.198 | 0.050 | 3.924 | **0.000** | Supported |
| H6 | PI –> GPPB | 0.514 | 0.051 | 10.136 | **0.000** | Supported |

EC: Environmental Concern, PI: Purchase intention, GPA: Green product Awareness, GS: Government support, PEV: Perceived ecological value, CGP: Community green practice, GPPB: Green product purchase behavior.

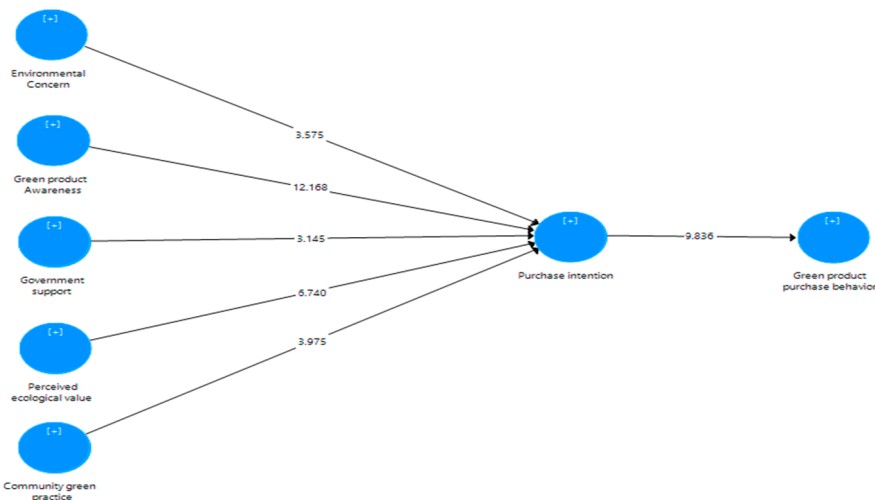

**Figure 4.** Structural model.

The research proposed six primary hypotheses to assess the relationship between the variables. According to Table 5, hypotheses H1, H2, H3, H4, H5, and H6 were accepted based on the empirical outcomes of this research. Hence, the study established a positive relationship shared by environmental concern, green product awareness, government support, perceived ecological value, community green practice, green product purchase intentions, and purchase intention towards green product purchase behavior.

### 4.2.5. Mediation Effect

Researchers have identified the use of bootstrapping for investigating the mediation effect in PLS-SEM as the more rigorous application available for inference statistics [66–69]. This study utilized the bootstrapping technique to obtain approximate t-values for significance testing of all path coefficients using 500 sub-samples. According to Table 6 below, purchase intention has a mediation effect between all factors towards green product purchase behavior and *p*-values are below 0.05.

**Table 6.** Mediation effect (indirect effect).

| | Mean, STDEV, t-Values, *p*-Values | | | | |
|---|---|---|---|---|---|
| Path | Original Sample (O) | Sample Mean (M) | Standard Deviation (STDEV) | T Statistics (\|O/STDEV\|) | *p* Values |
| Environmental Concern –> Purchase intention –> Green product purchase behavior | 0.073 | 0.075 | 0.025 | 2.903 | 0.004 |
| Government support –> Purchase intention –> Green product purchase behavior | 0.086 | 0.086 | 0.029 | 2.958 | 0.003 |
| Community green practice –> Purchase intention –> Green product purchase behavior | 0.102 | 0.103 | 0.028 | 3.639 | 0.000 |
| Green product Awareness –> Purchase intention –> Green product purchase behavior | 0.500 | 0.498 | 0.052 | 9.643 | 0.000 |
| Perceived ecological value –> Purchase intention –> Green product purchase behavior | −0.274 | −0.271 | 0.045 | 6.092 | 0.000 |

## 5. Discussion

In this research, more than 300 consumers participated in the questionnaire. All constructs have satisfactory reliability, with figures ranging from 0.803 to 0.863. The descriptive analysis and experiments of the PLS path model, which is the measurement model and structural model, were included in this research. The multicollinearity among independent variables ranged from 1.385 to 2.187 because the VIF value for independent variables fell below 5 and above 0.10, and the multicollinearity assumption was not violated. The fact that the multicollinearity assumption was fulfilled was also validated.

This study proposed six primary hypotheses to test the relationship between the variables. According to Hypothesis (H1), "There is a significant relationship between environmental concern and purchase intention towards green product purchase behavior." Table 5 shows the path coefficient for Environmental concern –> Purchase intention (EC –> PI) was 0.141, while the t-value was reported as 3.651 in the structural model's results. The hypothesis has a *p*-value of 0.000, higher than the value of t > 1.96, which has a *p*-value of <0.05. Hypothesis (H1) was accepted based on the above analysis. Such findings align with previous literature that shows a positive relationship between environmental concern and purchase intention towards green product purchase behavior [69]. Since ecological concern represents a vital element of environmental awareness components [70],

consumers have demonstrated their awareness of such matters and have increasingly altered their actions in favor of its defense (Shabbir et al., 2020). Hence, this analysis is also validated by the previous literature.

Hypothesis (H2) states, "There is a significant relationship between green product awareness GPA and purchase intention PI towards green product purchase behavior." According to Table 5, the path coefficient for GPA –> PI was 0.974, and the t-value 12.793. The structural model's results show the hypothesis's *p*-value is 0.000, more than the threshold value of t > 1.96, which has a *p*-value of < 0.05. Hypothesis H2 was accepted based on the above analysis. The results align with the previous study that green awareness is a crucial component created through green training and can be regarded as the dimension that drives education among consumers [71]. According to recent research [72], green awareness influences consumers' attitudes and acceptance of green products.

Hypothesis (H3) states, "There is a significant relationship between government support GS and purchase intention PI towards green product purchase behavior." Table 5 shows the path coefficient for GS –> PI was 0.167, while the t-value was reported as 3.208 in the structural model's results. The hypothesis has a *p*-value of 0.001, higher than the threshold value of t > 1.96 (*p*-value < 0.05). Hypothesis (H3) was accepted based on the above analysis. This analysis aligns with the previous studies. According to Adhitiya and Astuti [35], government policies play a crucial role in forecasting consumers' environmental attitudes, which means that the role of the government in protecting the environment positively impacts consumers' attitudes towards green goods. The government's support has a significant impact on the consumer consumption culture of market orientation through guiding firms to produce green and sustainable products and meet consumers' requirements [33,70].

Hypothesis (H4) states, "There is a significant relationship between perceived ecological value and purchase intention towards green product purchase behavior." Table 5 shows the path coefficient for PEV –> PI was −0.533, and the t-value was reported as 6.851, according to the structural model's results. The hypothesis has a *p*-value of 0.000, higher than the threshold value of t > 1.96 (*p*-value < 0.05). Hypothesis (H4) was accepted based on the above analysis. This finding aligns with prior research, which found that consumers are willing to pay more for environmentally friendly products and participate in sustainability initiatives [73]. According to the literature, modern consumers demonstrate more awareness of the environmental advantages and attributes of goods than previous generations, which, in turn, affects the perceived values of green products [37].

Hypothesis (H5) states, "There is a significant relationship between community green practice and purchase intention towards green product purchase behavior." Table 4 shows the path coefficient for CGP –> PI was 0.198, while the t-value was reported as 3.924 in the structural model's results. The hypothesis has a *p*-value of 0.000, higher than the threshold value of t > 1.96 (*p*-value < 0.05). Hypothesis (H5) was accepted based on the above analysis. This hypothesis aligns with the results of previous studies, which show consumers' willingness to spend more on green goods and participate in environmental practices, such as recycling, composting, and environment prevention [74]. Such consumers are the most aware of trustworthy, environmentally conscious green goods. According to previous research [75], it is crucial to adopt a more effective method of attracting consumers' attention and disseminating the value of green practices to them.

Hypothesis (H6) states, "There is a significant relationship between purchase intentions and green product purchase behavior." Table 4 shows the path coefficient for PI –> GPPB was 0.514, and the t-value was reported as 10.136, according to the structural model results. The hypothesis has a *p*-value of 0.000, higher than the threshold value of t > 1.96 (*p*-value < 0.05). Hypothesis (H6) was accepted based on the empirical findings of this research. Such a finding aligns with the empirical outcomes of the previous literature, which prove that the intention to buy green products represents the possibility that a consumer will purchase a specific product resulting from their environmental needs and

concerns [76]. Additionally, previous studies have demonstrated that attitudes, beliefs, and motivations play a crucial role as a predictor of green product purchasing intentions [42].

## 6. Research Implications and Contribution

This research contributes to the field of practice in sustainable consumerism and establishes awareness of purchasing green products by providing recommendations based on the examined factors and developed model. Firstly, from a practical perspective, the research findings offer integral building blocks applicable for marketers and firms to understand consumers' motivations to buy green products and participate in the environmental sustainability process. In other words, this study informs marketers, firms, and related government departments on appropriate strategies, allowing them to effectively engage consumers in the environmental sustainability process through green product purchasing decisions. Secondly, from a theoretical perspective, the research findings confirm that the factors influencing Malaysian consumers' intentions to buy green products are (a) product awareness, which plays a critical role in consumer decision-making processes when buying green products, followed by (b) perceived ecological value and (c) community green practices. These three factors, supplemented by environmental concern as a fourth factor, help foster ecological protection and a healthy society. Additionally, government policies should encourage such practices among consumers by implementing related regulations and redirecting the market and firms toward producing more affordable green products and participating in the environmental sustainability process.

Moreover, the findings of this research benefit all consumers who prioritize a clean and sustainable environment through purchasing green products. Such an approach benefits individuals and society and ensures that the environment will be cleaner and more secure in the future. The outcomes of this analysis will positively contribute towards creating awareness about green products, which will, in turn, influence the large-scale purchase of green goods and eventually help create a sustainable environment. Finally, the results of this research can be used as methods to encourage more consumers to purchase green products.

## 7. Conclusions

This research focuses on the different factors influencing consumers to buy green products. Marketers will use the outcomes of this analysis to determine the pattern of features that drive consumers to purchase green products. The results of this research can help the authorities foresee changes in environmental concerns. This research revealed that consumers show awareness of ecological threats and can safeguard and sustain their environment by purchasing green products, which will improve their likely future quality of life. Inflation in markets, on the other hand, might stop consumers from purchasing green products. The six factors of this research, which include environmental concern, green product awareness, government support, perceived ecological value, community green practice, and purchase intention, are significant factors that influence consumers to buy green products.

## 8. Limitations and Suggestions

Future studies should involve longitudinal research on consumer green purchasing behavior because such an approach can track patterns in consumer purchasing behavior and identify changes in consumer perception. Additionally, the results of this research may not remain accurate due to dynamic changes in consumer purchasing behavior. Longitudinal analysis is also more relevant for prospective investigations, according to researchers. Future research should concentrate on using different variables to validate the current research. Therefore, to learn more about the factors that influence consumers to buy a green product, the use of specific variables, such as different demographic factors, may have significant positive effects. Additionally, future research can use online questionnaires to generate a broader base of respondents.

**Author Contributions:** Data collection and analysis was conducted by N.H.A.-K., S.A. and F.M. The results and discussion were courtesy of M.S.S. and S.A.-S., while the methodology was the work of N.A.G. and A.K.A. Reviewing and editing was carried out by S.H.H., S.S.H. and H.M.A.A.-R. All authors have read and agreed to the published version of the manuscript.

**Funding:** This research received no external funding.

**Data Availability Statement:** The data presented in this study are available on request from the corresponding author.

**Acknowledgments:** The authors are grateful to the SuITE research group for guidance, CRIM for financial support, and Universiti Teknikal Malaysia Melaka for providing facilities in this study.

**Conflicts of Interest:** The authors declare no conflict of interest.

## Appendix A

**Table A1.** List of items in the questionnaire.

| Environment concern (EC) |
|---|
| Malaysia environment is one of my primary concerns. |
| I like to be involved in environmental protection activities performed in my country. |
| I often think about how to improve the environmental quality in Malaysia. |
| I support the idea of imposing anti-pollution regulations in my country. |

| Green product awareness (GPA) |
|---|
| Green products have no harmful effects on human health and the environment. |
| I encourage people who are important to me to choose and buy green products. |
| I have the time and desire to purchase green products. |
| I still want to purchase green products even at higher prices. |

| Government support (GS) |
|---|
| The Malaysian government endorses regulations that facilitate purchasing and using green products. |
| The Malaysian government is making an effort to set up facilities that allow the purchase of green products easily. |
| The Malaysian government encourages me to purchase green products. |
| I think the Malaysian government organises events/advertisements to attract citizens to purchase or use green products. |

| Perceived ecological value (PEV) |
|---|
| I prefer to keep the environment safe by using ecolabel products. |
| I prefer to keep the environment healthy and safe by avoiding buying products that harm nature. |
| I like to purchase products that reduce the disruption of nature. |
| I am ready to participate in any activity to reduce the impact of climate change. |

| Community green practice (CGP) |
|---|
| My community members participate in recycling paper and other recyclable materials. |
| My community members participate in energy-related conservation. |
| My community members participate in charitable activities and eliminate waste wisely. |
| My community has a reusable container for disposing of waste materials accordingly. |

| Purchase intention (PI) |
|---|
| I would like to consider purchasing green products first. |
| I plan to spend more on green products rather than conventional products. |
| I will consider purchasing green products because they are less polluting. |
| I expect to purchase green products in the future because of their positive environmental contribution. |

| Green product purchase behavior (GPPB) |
|---|
| Purchasing green products is a wise idea. |
| Purchasing green products would be pleasant. |
| I can decide whether to purchase green products or not by myself. |
| I am willing to spend resources and time and take any opportunities to use green products. |

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
