# Peer review of "Fostering a Clean and Sustainable Environment through Green Product Purchasing Behavior: Insights from Malaysian Consumers’ Perspective"

_sustainability, doi:10.3390/su132212585_

Round 1
Reviewer 1 Report
Dear Authors,
Congratulation for the well done paper and for the interesting results. The model is well built and data analysis shows robustness.
Some minor suggestions that would add value to the manuscript:
- introduction of additional information on the market of green products in Malaysia
- more details about the questions in the questionnaire (or attaching the questionnaire at the end of the paper)
- the way in which the questionnaire was applied and the period in which the study was carried out
Good luck!
Author Response
Many thanks to Reviewer1 inputs and comments
- Additional information on the market of green products in Malaysia already added supported with visualized figures.
- Details about the questions in the questionnaire already attached in the paper Appendix and the source of items already mentioned in the research methodology text.
- When the study was carried out? already updated in the methodology.
- Second professional proofread has been done using UK language

Reviewer 2 Report
Dear Author and Editor,
The paper is very on time and presents very interesting findings. The methodology is well written and the whole research project is very logical.
I have two small suggestions, that might help you to improve your manuscript.
First, try to build the background of the environmental situation in the world to illustrate the importance of the research.
Without action there will be no chance to reverse those changes or even more the ecosystem will be destroyed. Some interesting references about environmental changes you might find here:
https://doi.org/10.1080/00207233.2019.1644024
Also, this paper might be an interest to you as it shows the tendency:
https://doi.org/10.1111/jiec.13084
Both papers might help you to explain the current situation and show the forecast, which is important to your study.
Second, try to create a discussion section to visualize the results in light of what we already know (literature).
Author Response
Many thanks to Reviewer 2 for his/her comments and inputs, we have improved and updated the following thins:
- This reference already added https://doi.org/10.1080/00207233.2019.1644024
- Additional information on the market of green products in Malaysia already added supported with visualized figures.
- Details about the questions in the questionnaire already attached in the paper Appendix and the source of items already mentioned in the research methodology text.
- When the study was carried out? already updated in the methodology.
- Double professional proofread has been done using UK language

Reviewer 3 Report
The reviewed article concerns a very important topic, both from a theoretical and practical point of view, which is to know the motivation of consumers to purchase "green products".
The idea of the study, the method of obtaining empirical data, as well as the layout of the article are consistent with the principles of implementation and presentation of scientific results. Although the sample size may raise concerns, this issue has been explained and justified by the authors.
The method of data analysis is also correct and additionally the graphic presentation of the research concept deserves praise, facilitating the understanding of the authors' intentions and the essence of the study.
Information about the implementation of the survey is a bit unsatisfactory - it should be supplemented with data on the date of the survey (year) and the broadcasting technique used (CATI, CAWI?). It would also be useful to explain what the abbreviations EC1...EC4, GPA1...4 etc. stand for (Table 1). Explanation that these are items adapted from other surveys is insufficient.
The description of the results, their discussion and conclusion are not objectionable. However, it would be advisable to change the order of chapters 7 (Conclusion - why not Conclusions?) and 8 (Limitation and Suggestions - why not Limtations?).
I strongly suggest changing the form of literature references in the text, for example:
Line 53: is according to [6], and should be according to Chen et al. [6] (the same is true for lines 234-241, 384, and others). On the other hand, in line 384, a different way of referencing is used than required.
These minor shortcomings do not significantly diminish the value of the article.
Author Response
Many thanks to Reviewer for his/her comments and inputs, we have improved and updated the following thins:
- This reference already added https://doi.org/10.1080/00207233.2019.1644024
Additional information on the market of green products in Malaysia already added supported with visualized figures.
- Details about the questions in the questionnaire already attached in the paper Appendix and the source of items already mentioned in the research methodology text.
- When the study was carried out? already updated in the methodology.
- Double professional proofread has been done using UK language

Reviewer 4 Report
This study investigate the relationships among environmental concern, green product awareness, government support, perceived ecological value, green product purchase intention, and green purchase behavior. However, the study constructs were very commonly investigated in the many academic fields such as green marketing, consumer behavior and so no. Therefore, this study has somehow low originality as an academic paper. Above all else, this study has low quality of structure, clarification, logical coherence, strength of argument, and academic soundness. The detailed issues in this study are as follows.
- This manuscript needs a lot of English language editing. In addition, the logical connection between sentences is low. It is very hard to read.
Lines53-56)
-Additionally, according to [6], as several types of environmental damage are mostly caused by product materials, manufacturing processes should be minimized. Therefore, through the purchase of green items, consumers can help to prevent or mitigate environmental damage.
Lines67-74)
-Some users express a desire to buy green products to protect the environment but cancel the desire after entering the supermarket. This encourages major supermarkets such as Tesco, Aeon Big, and the Institute of Standards and Industrial Research Malaysia (SIRIM) to promote various types of green products such as Malaysian Consumer green electronic products, green packaging products, and various other types of green products to change the attitude of consumers to protect the nature [11].
Lines92-94)
-This shows that most friends and 92consumer families in Western countries are consumers of green products, thus, encourag- 93ing users to purchase green products. In Malaysia, most friends and family of consumers 94are not users of green products.
and so on..
I found lots of grammatical errors and weakness of logics in the statements in the rest parts of the manuscript as well. It seems not appropriate to use different terms as a same meaning in one manuscript such as consumer, customer, and user. And the way of citation is not right and the references contents are not match with the statements in the manuscript. In addition, many sentences don’t have appropriate references, (=> Lines 37-39/56-58/75-77, and so on) And I found there are many other issues that references didn't show names of journal in the reference list and some references were not written in English.
In the methodology, I couldn't judge whether the methodology and analysis are right or not as I couldn't find the measurement items in the manuscript. As the consumer’s purchase intention is generally used as a proxy variable of the consumer behavior in the marketing research, I think I should check how the authors measured these variables.
Finally, the research implications and contributions are very arbitrary. I couldn’t understand this research can conclude this research results lead fostering clean and sustainable environment through green product purchasing behavior.
Overall, I think this paper need lots of improvement in many aspects.
Author Response
Special thanks to this reviewer, really he/she gives many substantial comments to our study and we have updated and changed accordingly many of the points he/she recommended as followed:
- Double professional proofread already done using UK language and we paraphrased all statements that was commented.
- Additional information on the market of green products in Malaysia already added supported with visualized figures.
- Details about the questions in the questionnaire already attached in the paper Appendix and the source of items already mentioned in the research methodology text.
- When the study was carried out? already updated in the methodology.
- Double professional proofread has been done using UK language

Round 2
Reviewer 4 Report
I don't think this research has an enough improvement to be published in the journal.
It has no sufficient literature review to develop hypotheses, and the hypotheses had no originality to give any academic implications to green marketing field at all.
Therefore, I make a decision to reject this paper to publish in the journal.
Author Response
Dear reviewer 4
Hope you are fine
Thank you for your reviews, opinions and inputs
Herewith I included list of your comments and corrective action and improvements already taken as shown in the summarized table. Doubled proofread has been done to the entire manuscript.
Replying Reviewer 4 Comments
Reviewer 4 detailed comments:
|
Comments |
Corrective action taken |
Correction location |
1 |
Manuscript needs a lot of English language editing |
Manuscript has been proofreads by professional from UK. |
Copy of proofread Manuscript is attached separately as an evidence |
2 |
Lines53-56 very hard to read |
These lines already refined by adding correct referencing and rephrasing statement |
Colored in blue Lines 62-64 |
3 |
Lines67-74 ( wrong way referencing) and hard to read |
These lines already refined by adding correct referencing and rephrasing statement |
Colored in blue Lines 76-84 |
4 |
Lines92-94 very hard to read |
These lines have been removed and replaced with more related information from literature review |
Colored in blue Lines 101-114 |
5 |
Lines 37-39/56-58/75-77 |
These lines already refined by adding correct rephrasing statements |
New corrected lines |
6 |
Some references were not written in English |
Authors checked list of references and removed any reference not in English. |
List of references
|
7 |
Couldn't find the measurement items in the manuscript |
Measurement items already added |
See the Appendix |
8 |
Research implications and contributions are very arbitrary |
implications and contributions already refined and proofread |
Research implications and contributions |
|
|
|
|
